# Cord-Blood-Derived Professional Antigen-Presenting Cells: Functions and Applications in Current and Prospective Cell Therapies

**DOI:** 10.3390/ijms22115923

**Published:** 2021-05-31

**Authors:** Sarah Cunningham, Holger Hackstein

**Affiliations:** Department of Transfusion Medicine and Hemostaseology, University Hospital Erlangen, Schwabachanlage 12, 91054 Erlangen, Germany; holger.hackstein@uk-erlangen.de

**Keywords:** umbilical cord blood, antigen presentation, cell therapy, immune reconstitution

## Abstract

Human umbilical cord blood (UCB) represents a valuable source of hematopoietic stem cells, particularly for patients lacking a matching donor. UCB provides practical advantages, including a lower risk of graft-versus-host-disease and permissive human leukocyte antigen mismatching. These advantageous properties have so far been applied for stem cell, mesenchymal stromal cell, and chimeric antigen receptor T cell therapies. However, UCB-derived professional antigen-presenting cells are increasingly being utilized in the context of immune tolerance and regenerative therapy. Here, we review the cell-specific characteristics as well as recent advancements in UCB-based cell therapies focusing on dendritic cells, monocytes, B lymphocytes, innate lymphoid cells, and macrophages.

## 1. Introduction

Ever since the 1980s, umbilical cord blood (UCB) has been used as a rich source of allogeneic hematopoietic stem cells (HSC); these are primarily used for autologous pediatric transplantation or allogeneic transplantation in the absence of a fully human leucocyte antigen (HLA)-matched donor [1,2,3]. This is of particular importance to ethnic groups with low donation rates [3]. Major strides have been made to enhance UCB engraftment and reduce infectious risks, which have allowed for more than 40,000 UCB transplantations to this day. Worldwide, UCB from both public and private banks is being used to challenge neurologic, metabolic, immunologic, and neoplastic disorders.

The more recent surge in novel therapeutic technologies primarily focuses on the optimization of stem cell expansion and utilization of T lymphocytes and natural killer cells (NK) for chimeric antigen receptor (CAR)-based treatments. However, UCB cells have a rich and complex cell composition beyond the aforementioned cell subsets, each with their unique properties. Here, we review the immune cell composition of professional antigen-presenting cells (APC) in UCB and their clinical relevance to date.

## 2. APC Biology in Adult and Umbilical Cord Blood—What Is the Difference?

Numerous groups have demonstrated that UCB constitutes a rich source of HSCs. Likewise, early progenitors giving rise to hematopoietic colonies of different cell lineages, possessing long-term self-renewal properties, are more frequent in UCB samples in comparison with adult bone marrow (BM) or mobilized peripheral blood. Mouse studies also revealed that cord-blood-derived HSCs boast a higher hematopoietic reconstitution capacity in vivo in comparison with their adult BM counterpart [4,5]. In addition to HSCs, UCB also offers a rich source of mesenchymal stromal cells, which can differentiate into osteogenic, chondrogenic, and adipogenic cell lineages [6]. However, UCB also harbors numerous immunogenic subsets of the innate and adaptive immune system, which have become a unique and valuable source for cell therapies. The following section will give an overview of APC populations found in adult and neonatal blood (Figure 1).

### 2.1. Monocytes

#### 2.1.1. Monocytes in Adult Peripheral Blood

Monocytes represent critical effectors and regulators of the immune system. Equipped with chemokine receptors and toll-like receptors (TLRs), monocytes circulate in blood, BM, and the spleen under steady state conditions. Upon pathogen encounter, monocytes migrate from the blood to the affected tissue and produce inflammatory cytokines, including interleukine (IL)-12p70, IL-6, and IL-10, whilst also being phenotypically polarized by the microenvironment to mount specific functional programs [7]. In humans, three different monocyte subsets can be identified by the expression of CD14 and CD16, namely classical CD14^+^ CD16^−^, intermediate CD14^+^ CD16^+^, and non-classical CD14^−^ CD16^+^ monocytes [8]. Each subset possesses a distinct cytokine profile and differentiation potential. For instance, classical monocytes boast the highest capacity to secrete tumor necrosis factor (TNF) α, IL-6, and IL-1β, while non-classical monocytes are proficient at interferon (IFN) α secretion following TLR stimulation [8]. Furthermore, classical monocytes constitute the main source of monocyte-derived dendritic cells (MoDC) following IL-4 and granulocyte colony-stimulating factor (GM-CSF) incubation in vitro, while classical and non-classical monocytes fail to produce robust allogeneic T cell proliferation and IFNγ release [8]. Upon GM-CSF or macrophage colony-stimulating factor exposure, all monocyte subsets display macrophage (MΦ) morphology and cytokine profiles and enhanced phagocytosis, outlining the plasticity of monocyte differentiation. During homeostasis, monocyte subsets are thought to patrol the endothelium for damage and/or the presence of pathogens [9]. During an acute inflammation, fibroblasts, epithelial cells, and endothelial cells release C-C motif chemokine ligand 2 in response to inflammatory cytokines from tissue-resident MΦ and microbial antigens to allow for blood monocyte diapedesis [10]. This allows the predominantly CC chemokine receptor 2^+^ (CCR2) classical monocytes to transmigrate into the affected tissue. Following their entry, these cells are thought to act as end-type killer cells, as the non-specific toxic molecules they produce will also cause their own demise [11,12]. Non-classical and intermediate monocytes express only low levels of CCR2, while their high expression of C-X3-C motif chemokine receptor 1 allows them to patrol and for their prolonged survival [13]. Furthermore, non-classical and intermediate monocytes are weak phagocytes but secrete TNFα, IL-1β, and CCL3 in response to TLR7/8 stimulation. Generally, classical monocytes have a higher capacity than intermediate and non-classical monocytes to enter inflamed tissues and are able to convert into monocyte-derived MΦ with a pro-inflammatory M1-like phenotype.

#### 2.1.2. Neonatal Monocytes

In comparison with adult peripheral blood, UCB boasts higher frequencies and absolute counts of classical monocytes, while intermediate monocytes are slightly diminished (Table 1) [14]. Under steady state conditions, UCB monocytes express similar levels of CD11c, CD163, CD80, and HLA-DR in comparison to their adult counterparts [15]. Though several laboratories have studied the responsiveness of UCB monocytes to TLR agonists, the results do not always match. While some studies observe a high secretion of early response cytokines, such as IL-6, IL-8, IL-10, and IL-1β [15], others observe equal to lower secretion rates in comparison with adult monocytes following TLR stimulation [16]. The diverging results from these studies can be attributed to differences in the employed methods and modes of stimulation. While peptidoglycan-mediated TLR2 stimulation induces high levels of IL-12p70 and TNFα, these results can only be reproduced using Resiquimod (TLR7/8) [17,18,19], while lipopolysaccharide (LPS; TLR4) only induces low secretion levels [17,20,21,22]. UCB-derived monocytes primed with the Bacillus Calmette–Guérin vaccine display changes in IL-6, IL-10, and TNFα secretion reminiscent of trained immunity [23]. Similar to adult monocytes, UCB-derived monocytes show an altered cytokine profile following induced training with *Candida albicans* β-glucan, indicating that neonatal monocytes are able to maintain immunological information from previous antigenic experiences [23]. The underlying mechanism has been tied to the reprogramming of a Dectin-1/Raf1-dependent pathway and changes in the epigenetic profiles in monocytes [23,24]. This interplay has been proposed to explain the non-specific benefits of BCG vaccination observed in neonates [25]. Epigenetic studies addressing potential DNA alterations, such as methylation, histone modifications, and microRNA expression, are subjects of differential regulation in neonatal monocytes [26]. In particular, neonatal monocytes present an increase in methyl groups on lysine (K) 4 of histone (H) 3, e.g., H3K4me3 deposition on pro-inflammatory IL-1β, IL-6, IL-12β, and TNF cytokine promoters, which is associated with gene silencing through chromatin compaction [26].

Due to their diminished production of pro-inflammatory cytokines, neonatal monocytes have been suggested to act as a double-edged sword in the context of viral infections. One of the more recent examples can be seen in neonatal Zika virus (ZIKV) infections, whereby monocytes provide a potentially favorable environment for viruses. C-type lectin domain family 5 member A, a classical flavivirus immune receptor, is highly expressed in neonatal monocytes and acts as a viral entry receptor for ZIKV [27]. This, together with the lower expression rates of TLR3 and the low expression of pro-inflammatory cytokines, depicts neonatal monocytes as vessels for ZIKV, helping it to spread to the nervous system without triggering substantial immune responses [27,28]. Dreschers et al. were able to show that UCB monocytes displayed lower rates of phagocytosis-induced cell death due to lower TNF/TNFR1 internalization, resulting in lower apoptotic rates of effector cells [29]. This mechanism is associated with the antibacterial host response and contributes to a controlled resolution of inflammation [29,30]. Lastly, UCB-derived monocytes were shown to display impaired homeostatic extravasation in comparison with adult monocytes [31]. It appears that neonatal monocytes are intrinsically impaired with respect to traversing quiescent endothelia yet migrate through pre-activated endothelia at the same frequency as adult monocytes in a micro physiological model. This effect appears to be independent of autologous neonatal serum, as serum-free conditions did not alter neonatal CD16^+^ and CD16^−^ monocyte behaviors. Among the known constitutively expressed extravasation markers [32], CD31 and CD11b were found to be significantly downregulated in UCB CD14^+^ CD16^−^ and CD14^+/−^ CD16^−^ monocytes [31]. Altogether, these findings may explain the course of neonatal sepsis to some extent, yet further investigations are required. Still, neonatal monocytes present a valuable source material for the generation of DC and MΦ, which will be discussed in the following sections.

### 2.2. DC

#### 2.2.1. Adult DC

DC comprise a heterogeneous group of BM-derived cells arising from lympho-myeloid hematopoiesis that mediates the interaction between sensing and activation of both the innate and adaptive immune system [36]. DC constitute only 1% of human peripheral blood [37] and exist in both lymphoid and non-lymphoid tissues, serving as scavengers for pathogen- and danger-associated signals. DC possess the unique ability to process both extracellular and intracellular antigens, which are able to prime naïve T cells in the context of major histocompatibility complex (MHC) presentation. Based on their functional properties and interactions with T cells, DC are subdivided into three major subsets, including plasmacytoid DC (pDC) and two conventional (cDC) subsets called cDC1 and cDC2, defined by their CD141 and CD1c expression, respectively [38,39]. Both cDC subsets are able to drive CD4^+^ and CD8^+^ T cell stimulation via antigen presentation. CD141^+^ DC possess a high intrinsic capacity to cross-present antigens via MHC class I to activate CD8^+^ T cells and NK cells whilst producing comparably low levels of IL-12p70 [40,41,42]. CD1c^+^ DC, on the other hand, express a wide range of retinoic acid inducible gene I receptors, toll-like receptors (TLRs), lectins, and nucleotide-binding oligomerization domain (NOD)-like receptors similar to monocytes, producing high levels of IL-12p70, and can act as potent activators of CD8^+^ T cells and T_H_1, T_H_2, and T_H_17 populations [43,44,45,46]. pDC express low levels of MHC class II and CD11c while also being potent producers of type I IFNγ in response to viral infections [47,48]. These DC have a distinct lineage from monocyte-derived DC and MΦ populations. This flexibility is crucial to the immune system’s ability to react appropriately to a range of pathogens.

As mentioned above, DC can derive from monocytes (MoDC), which play a key role in inflammation. Although distinct from DC, MoDC play a complementary role to CD141^+^ and CD1c^+^ DC in the initiation and resolution of inflammation [36], acting as potent activators of T cells. Following CCR2^+^ C-X3-C motif chemokine receptor 1^low^ monocyte excavation into inflamed or infected tissue, T_H_1-associated cytokines favor inflammatory MoDC generation [49]. These MoDC phenotypically distinguish themselves from tissue-resident MΦ and monocyte-derived MΦ by their expression of CD11c^+^, MHCII^+^, CCR7^+^, CD11b^+^, CD205^+^, CD206^+^, and CD209^+^ and their lack of cDC-associated (CD1a, CD141) markers [50]. Following antigen uptake, processing, and presentation, MoDC secrete a range of cytokines, including IL-12p70, TNFα, IL-1β, and IL-23, to drive both CD4^+^ and CD8^+^ T cell activation [36]. In vitro, MoDC can be generated through a combination of GM-CSF and IL-4 to suppress MΦ development, followed by activation of stimuli including LPS and prostaglandin E_2_ [36]. The relative relationship between cDC and MoDC differs quantitatively and qualitatively, with monocytes acting as a reservoir of easily accessible immune modulators of the innate and adaptive immune system in case of an uncontrolled infection and/or inflammation to reinforce cDC function.

Further advances in gene expression studies have allowed for DC subset classification not only in terms of these functional properties, but also based on lineage, which correlates with the differential expression of transcription factors [36,51,52]. This development allowed for the distinction of six human blood DC subsets (DC1-DC6) using single-cell RNA sequencing [53]. These six subsets can be mapped to the already described DC subsets: DC1 corresponds to CD141^+^ DC [53]; CD1c^+^ DC can be split into DC2 and DC3 [54]; DC4 represents a CD141^−^CD1c^−^ double-negative subset associated with a monocyte-like cell type; DC5 corresponds to a newly defined AXL receptor tyrosine kinase^+^ DC subset that can act as a precursor for CD1c^+^ DC [53,55]; and pDC, which represent DC6 [53].

#### 2.2.2. Neonatal Dendritic Cells

The previously discussed DC subsets can be found in UCB; however, they differ in their frequency and function to adult peripheral blood [33,34,56]. In contrast to their adult counterparts, DC phenotypes are skewed toward immaturity, marked by reduced antigen uptake as a consequence of lower expression of the immunoglobulin (Ig) G receptors CD32 and CD64 [57] and lower expression of co-stimulatory molecules, including MHC class II, ICAM-1, CD80, and CD86 [58]. Furthermore, UCB-derived DC subsets secrete less TNFα, IL-12p70, and IFNγ following TLR stimulation [33]. Neonatal pDC stimulation with TLR4 (LPS) and TLR9 (CpG) leads to a markedly reduced production of IFNα in comparison with pDC from 1 year old infants and adults [59,60]. Exposure of neonatal cDC to LPS or Poly (I:C) leads to diminished upregulation of CD40 and CD80, while HLA-DR and CD86 expression does not differ significantly in comparison to adults [21]. This phenomenon of DC immaturity can to some degree be ascribed to the necessary prevention of the alloimmunization of the mother and fetus and the reduced antigenic load of the intrauterine environment. Functional studies concerning human UCB DC are scarce, particularly concerning discrete cDC subsets, with most of the knowledge stemming from murine studies. It is important to note that mice and humans display significant temporal differences in early life, particularly concerning the tissue distribution of DC subsets [61]. DC from mice and humans are subject to dynamic age-dependent changes after birth, with mice displaying a widespread scarcity of DC subsets until 5 weeks of age [62]. Interestingly, pre-treatment of murine cDC1 with GM-CSF, IL-4, and IFNγ boosted the capacity to produce IL-12 but did not reach adult levels. This implies that some pathogen sensing pathways are functional in neonatal cDC and can be enhanced through costimulatory signals to potentially boost immune responses in neonates. Murine neonatal cDC express lower basal levels of MHCII and CD80 in comparison with adult cDC, and both neonatal cDC1 and cDC2 are able to induce allogeneic CD4^+^ T cell proliferation similar to adult DC [62]. Furthermore, neonatal cDC1 cells avidly phagocytose and cross-present bacterial antigens to CD8^+^ T cells [63]. Still, neonatal murine neonatal cDC1 cells respond in both a pro-inflammatory and anti-inflammatory manner due to the additional secretion of IL-10 that suppresses CD8^+^ T cell activation [62]. Accordingly, IFNγ secretion by CD8^+^ T cells can be enhanced by blocking IL-10 using an α-IL-10R antibody [62]. Antigen exposure in neonates and infants can induce both T_H_1 and T_H_2 responses but predominantly induces a T_H_2 recall response later in life [61,64].

Studies addressing the lower incidence and severity of GvHD following allogenic UCB transplantation uncovered that neonatal DC express significantly higher amounts of the Fas ligand (FasL) in comparison with adult DC, leading to higher rates of apoptosis in co-cultured T cells [65]. UCB cDC expressing high levels of FasL are thought to limit the proliferation of T cells by killing activated Fas^+^ CD4^+^ T cells with T_H_1 displaying a higher sensitivity to FasL-mediated apoptosis than T_H_2 cells [66]. It has therefore been hypothesized that the higher frequency of FasL^+^ cDC in UCB and, therefore, the killing of activated CD4^+^ T_H_1 cells results in the polarization of T cells towards a T_H_2 phenotype associated with lower incidences of GvHD [65]. Taken together, these observations suggest that the tolerogenic properties of UCB DC subsets are the result of their diminished potency as professional APC and their apoptotic capability to regulate T cell responses. However, it remains to be clarified to what extent specific human UCB DC subsets contribute to this phenomenon.

As in adult peripheral blood, UCB monocytes can differentiate into DC (UCB-MoDC). Standard differentiation with GM-CSF and IL-4 produces a similar phenotype to adult MoDC, yet UCB-derived MoDC express lower levels of CD80, DC-specific intercellular adhesion molecule-3-grabbing non-integrin, and MHC molecules [67]. While IL-12p70 and TNFα secretion are diminished, IL-10 is potently expressed upon LPS and Poly (I:C) exposure or CD40 ligation [67,68]. The particular lack of IL-12p70 has been associated with a defect in the transcription of IL-12p35, which is the result of a decreased expression of IRF3-dependant genes in neonatal MoDC [33]. These findings imply that TRIF-dependent signaling pathways, leading to IFNβ synthesis, are hampered in UCB-derived MoDC and cannot be overcome by exposure to adult plasma, excluding the possibility of external factors, such as IL-10, being responsible for the observed effect [33]. However, NFκB-dependent pathways of the TRL4 signaling cascade are not affected in neonatal MoDC, as they produce comparable levels of TNFα, IL-6, and IL-8 in response to LPS [68]. Though neonatal MoDC present defects in T_H_1 T cell maturation and polarization, they are still able to induce T cell subsets. For instance, UCB-derived MoDC are able to induce melanoma antigen-specific CD8^+^ T cells, promoting IFNγ secretion and cytolytic activity towards tumor cells in vitro [69]. Various ligands, including the TLR2, TLR3, TLR4, and TLR8 ligands, are able to induce equal or higher amounts of IL-23 from UCB-derived MoDC, which is critical for the proliferation of T_H_17 T cells [33,70,71]. Additional reports unveiled that UCB-derived MoDC are able to induce Vγ9γδ T cell proliferation and cytotoxic behavior following generation with IFNα and GM-CSF [72]. Considering these observations, UCB-derived MoDC are generally thought to exhibit tolerogenic characteristics mediating the suppression of allograft rejection and supporting the survival of the fetus.

### 2.3. Macrophages

#### 2.3.1. Adult MΦ in Peripheral Blood

MΦ describe a heterogeneous group that is essentially omnipresent in tissues, taking on a range of functions, including both induction and resolution of inflammation, regeneration, and homeostasis. This dual nature is maintained in a delicate balance with phagocytosis contributing to inflammatory processes as well as innate and adaptive responses while the same processes restore homeostasis and promote tissue repair [73]. According to the current state of knowledge, most adult tissue MΦ have a non-monocytic origin. Under steady state conditions, BM-derived monocytes replenish tissue MΦ populations with high turn-over rates and serve as a reservoir for infiltrating MΦ following tissue injury and infection [74].

As discussed above, monocytes represent a heterogenous population, raising the question of whether specific subsets give rise to particular MΦ [75]. In short, MΦ have been historically classified into two categories in analogy to T_H_ nomenclature: type 1 (M1) activated MΦ and, alternatively, activated type 2 (M2) MΦ [76,77]. In vitro, M1 MΦ can be generated from monocytes using GM-CSF or macrophage colony-stimulating factor together with IFNγ in combination with LPS or TNFα, resembling a cytokine environment established by T_H_1 cells. M1 MΦ are associated with the release of nitric oxide and citrulline, enabling avid cytotoxic activity as well as other downstream toxic metabolites associated with the M1 “killing machinery” [12,78]. In contrast to M1 MΦ, M2 shift their arginine metabolism to produce ornithine and polyamines rather than NO in response to their specific stimuli [79]. In general, M2 cells participate in polarized T_H_2 responses, promote the killing of encapsulated parasites, possess high levels of scavenger, mannose, and galactose-type receptors, and are crucial to tissue remodeling and angiogenesis [80]. M2 MΦ represent a double-edged sword in terms of tumor growth, similar to M1 MΦ’s capacity to harm local tissues. As M2 describe all other forms of MΦ activation, four subsets have so far been identified: M2a, M2b, M2c, and M2d. M2a MΦ are induced by IL-4 or IL-13 and express high levels of CD206 and IL-1 receptor antagonists [81,82]. M2b MΦ can be generated following IL-1 receptor, TLR agonist, and immune complex stimulation and are marked by IL-1β, IL-6, IL-10, and TNFα secretion [81,82]. IL-10 and glucocorticoids are able to induce the M2c phenotype, which is able to promote strong anti-apoptotic responses through the secretion of high concentrations of IL-10 and tumor growth factor β (TGF β) [82,83,84]. Recently, a fourth population, M2d, has been identified, which is induced by TLR agonists in the presence of adenosine A_2A_ receptor agonists, promotes IL-10 and vascular endothelial growth factor secretion, and provides pro-angiogenic properties [81,85,86]. Interestingly, exposure of M1 MΦ to M2-associated stimuli and vice versa can to some extent lead to a re-programming of subsets [87], with molecular reprogramming of polarized MΦ being the subject of current research [88].

#### 2.3.2. Neonatal Macrophages

Reports addressing the function of MΦ from neonatal monocytes are rare. However, in a recent study, Drescher et al. note that neonates suffer more frequently from sepsis-related morbidity than adults, hinting at a differential immunological response towards invading pathogens [89]. MΦ polarization revealed that independently of the used stimulant, UCB-derived monocytes generated MΦ with lower expression of CD14, HLA-DR, CD86, and CD80, while similar expression levels of both IFNγ and IL-10 receptor could be observed [89]. Furthermore, neonatal UCB-derived MΦ present reduced expression levels of phagocytosis receptors and an alteration in glycolysis [89]. UCB-derived MΦ present a lower polarization capacity, a lower antigen presentation efficiency, and lower activation rates of specialized T cell subsets [90,91]. Typically, IL-10-activated adult MΦ overexpress scavenger receptors, such as CD163 and Fc receptors, which are essential to the phagocytosis of bacteria and debris; however, neonatal MΦ fail to do so [76,89]. Following activation, UCB-derived MΦ also express elevated levels of both IL-27 subunits inhibiting T_H_1, T_H_2, and T_H_17 lymphocytes [92]. Additionally, IL-27 can act in an autocrine manner on MΦ, inhibiting their response towards bacteria [93]. Interestingly, Dreschers et al. report that neonates display exceptionally high levels of S100A8/A9 alarmin proteins, which have been linked to the blockage of monocyte expansion and neonatal susceptibility to septic shock [89,94]. S100A8/A9 alarmins are secreted by neutrophils, display high bactericidal activity, and act as an endogenous TLR4 agonist [89]. It has been hypothesized that high levels of alarmins prevent hyperinflammation in order to allow commensal colonization of neonates in the first days after birth [89]. Exposure of neonatal MΦ to alarmins results in the shutdown of glycolysis and inhibition of the mechanistic target of the rapamycin (mTOR) pathway, which is critical to MΦ metabolism and activation. Furthermore, only UCB-derived MΦ displayed high expression levels of S100A8/A9 mRNA in comparison with adult MΦ [89]. Human HLA-G, a non-classical HLA class I molecule, has been reported to be involved in the immunomodulation of macrophages in the context of fetal–maternal tolerance and cancer [95,96,97]. HLA-G expression is restricted to a few healthy adult tissues and tumors with immunosuppressive microenvironments and poor therapeutic responses [96]. In neonatal MΦ, HLA-G may induce immunosuppression through engagement of inhibitory receptors, such as CD85j/immunoglobulin-like transcript 2, and polarizes MΦ toward a M2 phenotype inhibiting IFNγ secretion of T cells [97,98]. Cumulatively, these data demonstrate that UCB-derived MΦ display restricted immune responses marked by low T cell stimulatory capacities and predominant immunosuppressive responses.

### 2.4. B Lymphocytes

#### 2.4.1. Adult B Lymphocytes

B lymphocytes describe the center of the adaptive humoral immune system responsible for the production of antigen-specific Ig or antibodies against invasive pathogens. Each B cell expresses a unique antigen receptor, the B cell receptor (BCR), which constitutes a membrane-bound form of its antibody [99]. Recognition of an antigen by the BCR leads to reactive B cell differentiation and proliferation as well as the secretion of its particular antibody, which can be one of five Ig classes: IgM, IgD, IgG, IgA, or IgE [99]. In concert with follicular T_H_ (T_FH_) cells and DC, antigen-activated B cells undergo somatic hypermutation, a mutation rate ~10^6^ greater than the normal rate of mutation in the genome, of the variable V-(D)-J antibody gene segment to improve its antigen specificity and affinity [100,101].

B cells develop in specialized microenvironments of the fetal liver and BM from HSCs through a complex interplay of genetic programs and cytokine milieus. The major stages associated with B cell development include the HSC, the multipotent progenitor (MPP), and the common lymphoid progenitor, followed by the progenitor B cell (pro-B cell), the precursor B cell (pre-B cell), and the immature B cell [102]. For an in depth review of B cell development, see [102,103]. To initiate antigen-specific B cell activation and, therefore, a humoral immune response, B cells must encounter antigens via two core processes. First, lymphoid tissues are specialized in the filtration of body fluids and can present antigens to B cells. Second, blood lymphocytes are continuously recruited into antigen-rich compartments (lymphoid follicles) [104]. Here, a range of APC are able to take up antigens and present them to B cells, including specialized MΦ, follicular DC, and stromal cells efficient at presenting opsonized antigens [104]. Furthermore, antigen-engaged B cells can interact with cognate antigen-specific T cells that can be pre-activated by antigen-presenting DC. T_H_ cells have a profound influence on B cells through the expression of CD40L and the secretion of IL-4 and IL-21, supporting B cell proliferation and differentiation into plasma cells or germinal center B cells [104,105]. For most protein antigens, antibody secretion is dependent on the interaction between B cells and T_H_ cells. However, soluble proteins, haptens, bacterial polysaccharides, and LPS can stimulate B cells in the absence of T_H_. T independent antigens (TI) can be subdivided into TI-1 and TI-2 antigens, with TI-1 encompassing intrinsic B-cell-activating activity independent of the BCR specificity, whereas TI-2 initiate cross-linking, accumulation, and cross activation of BCRs in mature B cells [104,106]. Both TIs can lead to B cell proliferation and polyclonal as well as monoclonal antibody production [104,106,107]. The factors that determine whether B cells differentiate into plasma cells, germinal center cells, or memory cells is still being investigated. Recent studies associate high antigen affinity with strong plasma cell responses [108]. After their activation, B cells undergo isotype switching of their BCR from IgM/IgD to either IgG, IgA, or IgE. Typically, IgG represent the predominant isotype in serum followed by IgA, while mucosal secretions are dominated by IgA [104]. Memories of previously experienced pathogens can be maintained through long-term production of antibodies from long-lived plasma cells and by the formation of primarily resting memory B cells that can be reactivated upon antigen encounter [104,109,110]. Following re-exposure to a particular antigen, memory B cells rapidly differentiate into plasma cells and migrate to the BM to become long-lived plasma cells cyster2019. Memory, in terms of antibodies, is maintained in B cells of the IgM and IgG isotypes [104,111].

#### 2.4.2. Neonatal B Cells

The abrupt transition from a sterile intrauterine environment to a new one confronts the immune system of neonates with a challenging situation. Though different to their adult counterparts, neonatal innate immune cells are able to respond appropriately after birth, but are short lived and ineffective for prolonged immune protection [112]. Early immune protection relies on maternal IgG antibodies, which decline over a period of 2 months after birth [113]. The susceptibility of neonates to infections reflects both a general failure to generate T-cell-independent as well as T-cell-dependent B cell responses [112,114,115]. The human neonatal B cell repertoire is considered to be premature or impaired, similarly to most other APC subsets in UCB. UCB displays higher percentages and numbers of naïve B cells and transitional B cells than adults, while memory B cell populations are significantly reduced in neonates, as one expects [116]. In contrast, frequencies of isotype-switched and non-switched B cells slightly increase with increasing age [117]. In comparison with their adult counterparts, UCB-derived B cells display lower levels of surface molecules involved in B cell responses, including CD21, CD22, CD40, CD62L, CD80, CD86, and CCR7, decreasing their responsiveness to IL-10 and CD40L [118,119,120,121]. In addition, UCB B cell responses are influenced by external factors, including maternal antibodies clearing antigens in an epitope-specific manner and the immaturity of DC subsets to retain antigens to adequately stimulate B cells in an antigen-specific manner [112]. This consequently results in a decrease in somatic hypermutation, isotype class-switching in neonatal B cells, and lower antibody affinity [112,122]. It has also been hypothesized that immature DC–T cell interactions affect B cell IgG responses due to the preferential initiation of T_H_2 responses, either limiting or enhancing antibody responses, depending on the antigen-specific requirements of the specific B cell [112,123]. Neonatal CD27^+^ memory B cells, due to a lack of antigen exposure, are almost non-existent while CD27^−^ naïve B cells represent a predominantly transitional cell subset that has only recently emigrated from the BM and only contains a few mature naïve B cells [124]. The long-term production of antibodies requires the persistence of either memory B cells or long-lived plasma cells for adequate host protection. In infant mice, antibody production does not persist due to the failure of plasma cells to persist in the BM as long-lived plasma cells due to the lack of crucial differentiation and survival signals from BM-resident stromal cells [112,125]. The question of whether the same principle applies to human neonates remains to be answered, yet studies in human infant twins imply that the success of antibody production is only partially dependent on genetic determinants [126]. While the magnitude of antibody responses appears to be determined by genetic factors, the longevity of their production is dependent on environmental factors. This phenomenon has been theorized to explain the short-lived antibody responses in early life, with neonatal B cells competing for a limited set of plasma-cell survival niches within the BM following the newly experienced abundance of antigens after birth [112,126]. A further difference between neonatal and adult blood can be seen in the frequency of B1 B cells, which in contrast to conventional adaptive B2 cells secrete natural antibodies [127]. Natural antibodies possess lower affinity, recognize molecules on pathogens and self-antigens, and can be produced in the absence of exogenous stimuli [128]. Similarly to anergic B cells, B1 cells are relatively non-responsive to BCR engagement [129] and have been identified to display a CD20^+^CD27^+^CD43^+^CD38^lo/int^ phenotype [128,130]. B1 cells efficiently present antigens and are able to stimulate surrounding T cells with CD80/86 while also being able to act in an immunosuppressive manner by producing IL-10 [131,132]. It has been shown that with increasing age B1 populations and protective IgM natural antibody levels decline, in part explaining the higher susceptibility of the elderly to encapsulated pathogens [128]. The existence of B1 cells in humans has been the subject of many debates due to the difficulty of accessing sites in humans, such as the peritoneal cavity, where B1 cells are enriched in mice [132,133,134]. The properties associated with B1 responses include, similarly to mouse B1 cells, the ability to spontaneously secrete IgM antibodies that are broadly reactive, autoreactive, and repertoire-skewed, and the ability to readily prime T cells [134]. The probability that human B1 cells possess either of these described abilities, or both, depends on the expression of CD11b. Human CD11b^−^ B1 cells tend to be more focused on the expression of IgM while CD11b^+^ B1 cells appear to be more reactive to T cells [134,135]. However, further studies are required to verify if the previously identified and defined murine B1 subsets are also present in human UCB. Another difference between neonatal and adult blood B cells can be seen in the frequency of CD19^+^CD24^hi^CD38^hi^ regulatory B cells [136]. While regulatory B cells represent only 4% of the B cell compartments in adult peripheral blood, nearly 50% of UCB B cells present a regulatory phenotype, which declines with increasing age [137]. Upon stimulation with CD40L, CpG, or BCR ligation, regulatory B cells produce large amounts of IL-10, suppressing the release of IFNγ and IL-4 from T cells, and are able to mediate cell–cell contact-dependent suppression involving CD80, CD86, and CTLA-4 interactions [137]. These particular features have been linked to the lower rates of GvHD following UCB transplantation and have become new vistas in the treatment of transplantation-related complications [137].

### 2.5. Innate Lymphoid Cells

#### 2.5.1. Adult Innate Lymphoid Cells

Innate lymphoid cells (ILCs) describe a recently defined family of immune cells that mirror the phenotypes and functions of T cells [138]. In contrast to T cells, ILCs lack recombination activating gene (RAG)-dependent rearranged antigen receptors and do not undergo clonal selection or expansion [138]. Rather, ILCs are able to respond to signals of infected or injured tissues and direct the early immune response in accordance with the initial insult [138]. ILCs are primarily found in lymphoid and non-lymphoid tissues, particularly on mucosal surfaces. However, ILCs can also be identified as a rare subset in peripheral blood and have been shown to possess antigen-presenting capabilities. How ILCs interact with the microenvironment and both arms of the immune system is the subject of ongoing studies and holds great potential for both basic research as well as preventive and acute cell therapies.

ILCs derive from the BM-situated common lymphoid progenitor and can be split into three groups based on their phenotype and secreted molecules: group 1 ILCs (NK and ILC1 cells), group 2 ILCs (ILC2 cells), and group 3 ILCs (ILC3 cells and lymphoid tissue inducer cells) [139,140]. Group 1 ILCs are defined by their release of IFNγ and the lack of T_H_2-cell- and T_H_17-cell-associated cytokines [140]. The prototypical member of group 1 ILCs are NK cells, which possess cytotoxic capabilities, are potent producers of IFNγ and defined as CD16^+^, CD56^+^, and CD94^+^ in humans, and partially overlap with other ILC subsets [140]. ILC1 cells are marked by a lack of CD117 and high expression of T-bet, while IFNγ production and development of ILC1 and NK cells in the presence of IL-12 and IL-18 are associated with a downregulation of the transcription factor retinoic acid receptor-related orphan receptor gamma t (RORγt) [140]. Group 2 ILCs are associated with IL-7-dependent development and secrete T_H_2 cytokines, such as IL-5, IL-6, IL-9, and IL-13, in response to IL-25 and IL-33 [140,141,142]. In particular, ILC2s are important in the defense against nematodes and respiratory tissue repair mediated by the release of the epidermal growth factor family member amphiregulin [143]. ILC2s are able to take up and process antigens, followed by MHC-II-associated antigen presentation and the expression of CD80 and CD86, to initiate antigen-specific T cell expansion to an equivalent level to that of pDC and naïve B cells [144,145]. Human ILC2s also express MHC II and are able to present antigens, which indicates that this particular feature might be conserved. Human ILC2s can be identified by their expression of receptor subunits of IL-33 and IL-25, CD161, and prostaglandin D2 receptor 2 [140]. The secretion of IL-17A and IL22 and the dependence on the transcription factors RORγt and IL-7Rα define group 3 ILCs [140]. ILC3s are similar to lymphoid tissue inducer cells, in terms of their dependency on RORγt, yet differ in their cytokine profile and marker expression. ILC3s encompass a heterogeneous population expressing the natural cytotoxicity receptors NKp30, NKp44, and NKp46 and can produce IL-22, IFNγ, and IL-17A [140]. Furthermore, ILC3s display antigen presentation capabilities, yet lack the expression of the co-stimulatory molecules necessary for T cell responses [146].

Though scarce, the discussed ILC subsets can also be identified in healthy peripheral pediatric and adult blood through strenuous gating strategies involving the exclusion of cell-surface markers identifying other immune cells [35,147]. Still, ILCs have been shown to be involved both in promoting and suppressing tumor growth, highlighting both the need for further investigations and the potential for cell therapy approaches [146].

#### 2.5.2. Neonatal Innate Lymphoid Cells

The search for ILC subsets and their progenitors has only recently expanded to UCB using RNA sequencing and flow cytometric approaches. So far, only three studies have addressed the phenotypical characterization and functional properties of ILCs in human UCB, unveiling distinct differences between adult and neonatal blood. Frequencies of ILCs are significantly higher in UCB compared with adult and pediatric peripheral blood [35]. Interestingly, the number of ILC1-like cells rapidly decreases with increasing gestation and are thereafter found at consistent levels throughout life [148]. Unlike ILC1 cells, ILC2 and ILC3 cell subsets rapidly decrease following birth [148]. This implies that circulating ILCs migrate into tissues and become part of the local innate immune system following birth, which is supported by the differential expression patterns of DNA binding inhibitor 3 and 2 [148]. Both are known to be essential to the development of ILC subsets as well as their immunological properties [148]. Further in-detail analysis of UCB ILCs revealed that neonatal ILC1s express CR2, CD27, CD9, and CD200, while ILC2s and ILC3s express the receptor for CD27, CD70, which hints at cellular crosstalk between the respective cell subsets [148]. Furthermore UCB-derived ILC1s were shown to act as progenitor cells for NK cells [149]. These ILC1-like NK progenitor cells are marked by the expression of T-cell-specific molecules such as CD2 and CCR4. While present in adult peripheral blood, they possess a significantly reduced potential to produce mature NK cells [149]. Stimulation responses to known ILC-inducing cytokines, including IFNγ, IL-5, and IL-17A, showed that in comparison to known adult responses, only ILC2s exert effector functions [148]. ILC3s displayed a hampered response in terms of IL-2 stimulation yet acquired effector functions following TLR2:1 stimulation with Pam_3_CSK_4_ [148]. So far, circulating ILC3 cells have been depicted as functional inert cells, yet these findings hint at a particularly crucial role during pregnancy. ILC3s are among the innate immune cells that can traverse the placental barrier and may be vital in the defense against *Toxoplasma gondii*, which can be recognized by TLR2:1 [148,150]. In the presence of pregnancy-related factors and hormones in vivo in mice and in vitro in humans, ILCs display a reduced antigen presentation potential, evidenced by the lower expression of MHC II following inflammation [151]. This indicates that, during pregnancy, ILC functionality, in terms of antigen presentation, is redirected towards tolerance to support pregnancy [151]. Currently, no data are available on the antigen presentation capabilities of UCB ILC populations and further studies by multiple laboratories are needed to identify the therapeutic potential of UCB-derived ILCs.

## 3. Current and Prospective UCB-Based APC Cellular Therapies

Cell therapies based on UCB have gained increasing momentum in patient care, primarily in the treatment of pediatric hematological malignancies but also in oncology and immune reconstitution applications. In addition to current clinical applications, much effort is put into the investigation of potential approaches in pathophysiology, including muscle regeneration, myocardic ischemia, and myeloablation [152]. Though mostly focused on HSC and MSC transplantation, recent studies have suggested that the beneficial effects in tissue regeneration can be attributed to UCB-derived subpopulations, particularly APC subsets that possess immunomodulatory functions. Mononuclear cells from human UCB have been successfully employed for the treatment of cerebral palsy [153], hypoxic–ischemic encephalopathy [154], and acute ischemic stroke [155] in recent years, reaching Phase I and Phase II clinical trials [156,157]. The clinical benefits have been attributed to the release of neurotrophic and anti-inflammatory factors that, e.g., support brain tissue repair [156]. Clinical human trials involving UCB-derived APC cell products remain rare, with one listed ongoing Phase I clinical trial employing UCB-derived monocytes for the treatment of acute ischemic stroke (NCT02433509). Still, numerous pre-clinical trials are investigating the clinical feasibility and effectiveness of UCB-derived monocytes, MΦ, and DC (Table 2). Though cellular therapies are starting to employ the prowess of B cells and ILCs in antitumor immunity [158,159], so far none have employed UCB to do so. The following section will provide insights into current and prospective cell therapies harnessing UCB-derived monocytes, MΦ, and DC in both mice and humans.

### 3.1. UCB-Derived Monocytes in Cell Therapies

Cellular therapies harnessing the potential benefits of UCB-derived monocytes have so far been focused on tissue recovery and repair, particularly involving disorders of the central nervous system. Saha and colleagues investigated the neuroprotective properties of UCB-derived CD14^+^ monocytes in mice in the context of oxygen and glucose deprivation following hypoxic–ischemic injury [156]. The observed neuroprotective capabilities were attributed to the release of tissue-repair-associated soluble factors, including chitinase 3-like protein-1, inhibin- A, IL-10, matrix metalloproteinase-9, and thrombospondin-1, which are overexpressed in UCB-derived CD14^+^ monocytes in contrast to adult monocytes [156]. A similar effect was observed in a middle cerebral artery occlusion rat stroke model [171]. Intravenous administration of mononuclear UCB cells, including CD14^+^ monocytes, 48 h post-stroke led to a significant reduction in infarct size and promoted functional recovery in contrast to CD14^+^-monocyte-depleted controls [171]. While early inflammatory responses, including microglia proliferation and neuroprotective insulin-like growth factor 1 secretion, were not affected in the discussed model, the secondary inflammatory response, associated with a significant influx of MΦ and pro-inflammatory cytokines, was affected by the introduction of UCB-derived mononuclear cells [171]. The exact mechanism of UCB-derived CD14^+^-monocyte-mediated neuroprotection remains to be elucidated. However, it has been implied that UCB monocytes tend to take on M2 MΦ-associated alternative activated phenotypes that in turn induce alternative activation of local myeloid cells [171,172]. Interestingly, another study investigating the potential neuroprotective abilities of UCB subsets in a hypoxic–ischemic murine brain injury model were able to identify suppression of peripheral blood mononuclear cell (PBMC) proliferation in vitro as well as T_H_1:T_H_2 and T_H_17:T_H_2 ratios in vivo following UCB-derived CD14^+^ monocyte injection [160]. However, UCB-derived CD14^+^ monocyte administration lead to an increase in neuro-inflammatory responses and an increase in T_H_ subsets in treated mice [160]. The authors argue that the CD14^+^ compartment contains both monocyte-derived suppressor cells as well as other, potentially activated, monocyte subsets, which were both removed in the previous model [160]. The question of whether a particular UCB monocyte subset is responsible for the observed neuroprotective benefits has not been investigated so far and requires further in-depth analysis.

Other beneficial effects of UCB-derived monocytes can be seen in novel cell therapeutic approaches to the treatment of Alzheimer’s disease. Though pharmacological targets have been discovered in recent years, disease-modifying treatments are still in the making. Initial experimental approaches using low-dose infusions of UCB cells were able to ameliorate cognitive impairments and Amyloid-β plaque neuropathology in amyloid precursor protein/presenilin 1 transgenic mice [162]. In a follow-up study, the group was able to identify UCB-derived monocytes, not adult monocytes, as the central players in the observed cognitive and Amyloid-β plaque-ameliorating benefits [161]. In general, Amyloid-β accumulation is associated with an influx of microglia, monocytes, and pro-inflammatory cytokines inducing localized inflammation [161]. Both short-term (2 months) and long-term (4 months) administration of UCB-derived monocytes reduced Amyloid-β plaque-associated pathology as well as soluble and insoluble Amyloid-β levels in the brain [161]. This phenomenon has been linked to the surface expression of the amyloid precursor protein metabolite soluble amyloid precursor protein α (sAPPα) on monocytes, which enables Amyloid-β binding. It has been implied that Amyloid-β/sAPPα heterodimerization leads to phagocytosis and degradation of Amyloid-β. Furthermore, it has been shown that UCB serum significantly impacts the expression of sAPPα, with a CI complement acting in a dose-dependent manner on sAPPα production [173]. Interestingly, older or adult monocytes express lower levels of sAPPα than UCB-derived monocytes, indicating that UCB-derived monocytes could provide new therapeutic options for the treatment of Alzheimer’s disease [161]. A subsequent study focusing on the therapeutic effects of UCB cells in a vascular dementia multiple microinfarction rat model hints at a similar benefit [174]. Following intravenous UCB cell administration, short-term and long-term memory amelioration and axon and myelin density increased and glymphatic function improved [174]. However, it remains to be determined if UCB-derived monocytes are responsible for the observed beneficial effects within this particular model.

In an in vitro model, Anh and colleagues studied the effects of IFNα-induced UCB-MoDC on allogenic adult T cells [72]. Here, IFNα-induced UCB-MoDC induced a 46.1 + 9.3-fold increase in Vγ9γδ T cells with CD4^+^ and CD8^+^ T cell numbers declining after 7 days of co-culture, similarly to conditioned media from IFNα-induced UCB-MoDC. In addition, IFNα-induced UCB-MoDC enhanced the cytotoxic capabilities of allogenic Vγ9γδ T cells in terms of A549 tumor cell lysis [72]. Previous studies were able to show that Vγ9γδ T cells halted tumor growth in vivo and in vitro by reducing the activation of pro-survival molecules and increasing the levels of apoptotic molecules, including the AKT and ERK signaling pathways [170]. With Vγ9γδ T cells accounting for only 5% of peripheral blood T cells, IFNγ-induced UCB-MoDC offer a new therapeutic way to specifically generate Vγ9γδ T cells for antitumor cellular therapies.

### 3.2. UCB-Derived MΦ in Cell Therapies

MΦ represent crucial components of the innate immune system due to their phagocytic activity, antigen presentation, and flexible phenotypes in mice and humans [175]. Similarly to monocytes, UCB-derived MΦ are being studied in the context of central nervous system malignancies, such as the demyelination of neurons. Proposed causes of demyelination include genetic and environmental factors, which can manifest as multiple sclerosis, spinal cord injury, or leukodystrophies [166]. Microglia and monocytes are crucial regulators of brain remyelination, participating in both the propagation and resolution of central nervous system injuries, debris clearance, and support of local oligodendrocytes to remyelinate local neurons. Saha et al. developed a cellular product, termed DUOC-01, on the basis of UCB-derived CD14^+^ monocytes, possessing properties from both MΦ and microglia [166]. As a proof of concept, the ability of CD14^+^ monocytes and DUOC-01 cells were evaluated in terms of remyelination of the brains of NOD/severe combined immunodeficient (SCID)/IL-2Rγ^null^ mice following cuprizone-feeding-mediated demyelination [166]. Following intracranial injection of DUOC-01, mice displayed enhanced myelination in the corpus callosum region, proliferation of oligodendrocytes, and lower rates of gliosis and cellular infiltration of microglia and astrocytes [166]. These findings indicate that while UCB-derived CD14^+^ monocytes accelerate remyelination, they do so to a significantly lower degree than DUOC-01 cells due to transcriptional changes initiated by the manufacturing process of the latter [166,167,176]. The questions of whether DUOC-01 cells persist in an immune-competent individual and what clinical impacts the low immunogenic reaction, as seen in mixed-lymphocyte assays, has on patients remain to be investigated [166]. However, studies addressing the therapeutic potential of UCB-derived MΦ remain rare.

Recent developments in MΦ cellular therapies will potentially soon harness the benefits of UCB and potentially affect the outcome of neonatal sepsis as well as novel strategies for immunotherapies. One approach can be seen in the “education” of MΦ to either enhance or mitigate inflammatory immune responses [177]. In a recent study, MΦ were genetically modified using CRISPR/Cas9-mediated Nod-Like receptor family pyrin domain containing 3 (NLRP3) ablation via a cationic-lipid-assisted nanoparticle to suppress NLPR3-associated inflammasome responses in the context of septic shock [177]. NLRP3 describes a cytosolic sensor of exogenous pathogens that upon activation ultimately results in the release of numerous pro-inflammatory cytokines, including IL-1β and IL-18. The NLRP3 inflammasome has been associated with a set of inflammatory diseases with successful inhibition being effective in the attenuation of septic shock [178], Alzheimer’s disease [179], atherosclerosis [180], and gout [177,181]. The blockage of NLRP3 prevented both LPS-induced septic shock as well as monosodium urate crystal-induced peritonitis in mice, as seen by significantly reduced serum levels of IL-1β, caspase-1 p10, and IL-18 [177]. However, blockage of MΦ NLPR3 did not entirely prevent septic shock or peritonitis in mice, most likely because of the simultaneous activation of other inflammasomes, such as NLRP1 and AIM2 [182]. Still, genetic modification of MΦ presents a promising therapy for NLRP3-dependant acute and chronic inflammatory diseases [177]. It remains to be determined whether this method is applicable for the treatment of neonatal septic shock as well as UCB-derived MΦ allogenic cellular therapies.

Novel advances in MΦ-based cell therapeutics can also be seen in the development of human chimeric antigen receptor (CAR) MΦ for cancer immunotherapy [183]. CAR T cell therapies have shown rapid advancements from promising pre-clinical models in oncology to commercial approvals for leukemia and lymphoma therapies [184]. However, their application to solid tumors has been difficult as a consequence of difficulties involving T cell trafficking into solid tumors [185]. Due to their ability to penetrate solid tumor tissue [186] and particular effector functions, a recent study generated the first CAR-MΦ for cancer immunotherapy [183]. Here, CAR-MΦ were generated using a chimeric replication-deficient adenovirus carrying an anti-human epidermal growth factor receptor 2 CAR. CAR-MΦ displayed a durable M1 phenotype that remained stable upon M2 IL-4, IL-10, and IL-13 stimulation as well as tumor-conditioned media exposure [183,187]. Within two humanized NOD-SCID-IL2Rγ_null_-3/GM/SF xenograft mouse models, CAR-MΦ were able to maintain the M1 phenotype, while control-donor-matched MΦ were converted to M2. Furthermore, CAR-MΦ presented antigen-specific phagocytosis, were able to stimulate local M2 MΦ to switch to a M1 phenotype, were able to recruit and stimulate CD3^+^ T cells to tumor sites, and significantly increased survival rates [183]. The question of whether the presented approaches can be applied to neonatal MΦ, harnessing the benefit of UCB availability and more permissive HLA matching, for allogenic cell therapies remains to be tested.

### 3.3. UCB-Derived Dendritic Cells in Cell Therapies

DC are key elements in modern cellular therapies with applications harnessing their multifaceted prowess as cellular vaccines in the treatment of, e.g., malignant cancers [188]. For instance, UCB-derived DC have been used in pilot phases of clinical trials of hematological disorders, including acute myeloid leukemia (AML) in pediatric patients [163,165,189]. Pediatric AML patients present low survival rates due to relapse even after last-resort hematopoietic stem cell transplantation [163,164]. Since UCB presents lower risks of GvHD as well as enhanced Graft-versus-leukemia, a pilot study investigated the potential benefits of a synchronized stem cell transplantation and DC-based vaccine from the same UCB donation [163,190]. Here, DC are generated from UCB-derived CD34^+^ cells in order to enhance the rapid generation of tumor-specific cytotoxic T cell responses and enhanced Graft-versus-leukemia in vivo [163]. Because UCB contains limited amounts of mature DC, CD34^+^ cells were selected, expanded, and differentiated into Wilms Tumor 1 (WT1) antigen-carrying UCB-derived DC under good manufacturing practice (GMP) conditions [163]. Within this pilot study, UCB-derived DC carrying the WT1 antigen presented upregulation of CD209, HLA-DR, CD83, CCR7, and CD11c, migratory behavior following a CCL19 chemokine gradient, as well as CD4^+^ and CD8^+^ expansion in an allogeneic MLR setting [163]. In their follow-up study, Plantinga and colleagues presented a GMP-compliant bag cultivation strategy for the production of the UCB-derived WT1 antigen-presenting DC vaccine [164]. The so-called CBDC vaccine is again generated from UCB-derived CD34^+^ cells, which are differentiated with Flt3-L, SCF, GM-CSF, and IL-4 followed by exposure to a gold-standard maturation cocktail, a WT1 peptivator, as well as WT1 mRNA electroporation [164]. The CBDC vaccine expresses CD11c, HLA-DR, CD83, and CCR7 and is able to induce both potent WT1-specific and AML-specific cell lysis, forming the basis for subsequent clinical trials [164]. In a similar approach, UCB-derived CD34^+^ cells are used to generate a Langerhans cell-like dendritic cell subset-based vaccine for the treatment of adult AML [165]. Expansion of UCB-derived CD34^+^ and subsequent exposure to either a combination of GM-CSF and IL-4 or TGFβ1 were used to generate clinically relevant numbers of mature CD1a^+^ myeloid DC and CD207^+^ Langerhans cells, respectively [165]. Both subsets produced minor histocompatibility antigen 1 and 2-specific cytotoxic T cell responses in the peripheral blood of adult leukemia patients in vitro and in a humanized NOD-SCID-IL-2Rγ^−/−^ mouse model [165]. Both UCB-derived Langerhans-like and myeloid DC-like cells display avid expression of CD11c, HLA-DR, CD40, CD83, and CD86 while favoring T_H_1-associated IL-12β secretion over T_H_2 and pro-inflammatory cytokine responses [165]. These preclinical findings still require the application of GMP-compliant production protocols yet open up the vista for specialized Langerhans cell and CD1a^+^ myeloid DC-mediated cell therapies in adults.

In a different approach, UCB-derived DC are being employed as effector cells for the development of cytokine-induced killer cells (CIK) [168]. CIKs describe a heterogeneous population of ex vivo expanded T cells that acquire CD56 expression during their expansion and boast the capabilities of both T and NK cells [191]. In contrast to other adoptive cell therapies, CIKs can be relatively easily obtained through cytokine stimulation of PBMCs or UCB cells and do not require antigen-specific stimuli to recognize and act on tumor cells [191]. Co-culture of UCB-derived DC and CIKs was shown to produce an almost 27-fold increase in CIK proliferation in comparison with CIKs by themselves and be superior to a co-culture with adult peripheral blood-derived DC [168]. Furthermore, UCB-derived DC–CIK co-cultures led to an increase in cytotoxic activity against several acute leukemia cell lines and supported the secretion of IFNγ, TNFα, and IL-12 [168]. However, it remains to be elucidated whether the co-culture also provides beneficial effects on the functionality of UCB-derived DC that in turn may positively affect the toleration and effectiveness of the envisioned cellular therapy.

In a recent study, UCB-derived DC were exploited for their production of exosomes and application in anti-tumor immunity [169]. Exosomes describe nanometric vesicles that are filled with a range of biomolecules and have been shown to be released into the extracellular environment by human primary MoDC and DC cell lines [169,192]. Furthermore, exosomes can also present antigens to T cells due to the presence of MHC II molecules as well as other co-stimulatory molecules on their surface [193] and have been reported to mediate tumor rejection in several clinical trials [169,194,195]. Interestingly, exosomes extracted from allogenic UCB-derived and differentiated MoDC induced potent CD8^+^ T cell stimulation in an in vitro lung cancer model [169]. Exosomes were harvested after successful generation of UCB-derived MoDC and pulsed with the A549 lung carcinoma tumor cell lysate expressing the exosome-associated markers CD9 and CD63 [169]. It is notable that proliferation of CD3^+^Vγ9 and CD3^+^CD8^+^ but not CD3^+^CD4^+^ T cells was induced by pulsed UCB-derived MoDC as well as a combination of UCB-derived MoDC and exosomes [169]. While it has been reported that pulsed MoDC are able to produce exosomes with pro-inflammatory or tolerogenic properties, the latter has so far not been addressed in UCB-derived MoDC and DC subsets. Interestingly, while exosomes from un-pulsed UCB-derived MoDC could not initiate allogenic T cell cytotoxicity, they were still able to prime allogenic PBMCs to efficiently kill A549 cancer cells in vitro [169]. Pre-clinical studies have established that cryopreserved UCB provides a readily available and valuable source of allogenic DC, particularly for cancer patients with limited DC. Due to the lower risk of GvHD, several pre-existing DC vaccine protocols have been and will continue to be applied to cryopreserved UCB to generate pro-inflammatory-oriented DC. Still, it remains to be investigated whether the tolerogenic potential of UCB-derived MoDC and DC can be efficiently harnessed for cellular therapies, especially for autoimmune disorders.

## 4. Conclusions

Widespread use of UCB as a source of HSCs and mesenchymal stromal cells for the treatment of leukemia or inherent metabolic diseases has been complemented by the exploitation of UCB-derived immune cells in regenerative therapy. Due to pre-clinical studies, UCB has been recognized as a feasible and standardized cryopreserved source of regenerative cells. UCB offers several advantages over other sources of HSC and immune cell subsets, including a lower rate of chronic GvHD in the case of a higher HLA disparity and the benefit of being an “off-the-shelf” resource that can be collected painlessly. However, depending on the desired cell population, limitations in cell dosage and delayed engraftment remain the main challenges in UCB-based cellular therapies. This bottleneck requires efficient ex vivo expansion strategies to enable HSC multiplication or the expansion of clonogenic cell subsets.

In summary, UCB provides an invaluable source of APC for the field of regenerative medicine. The exploration and development of Phase I–III clinical trials to evaluate the potential of UCB-derived APC populations remains an essential goal in translational research and medicine.

## Figures and Tables

**Figure 1 ijms-22-05923-f001:**
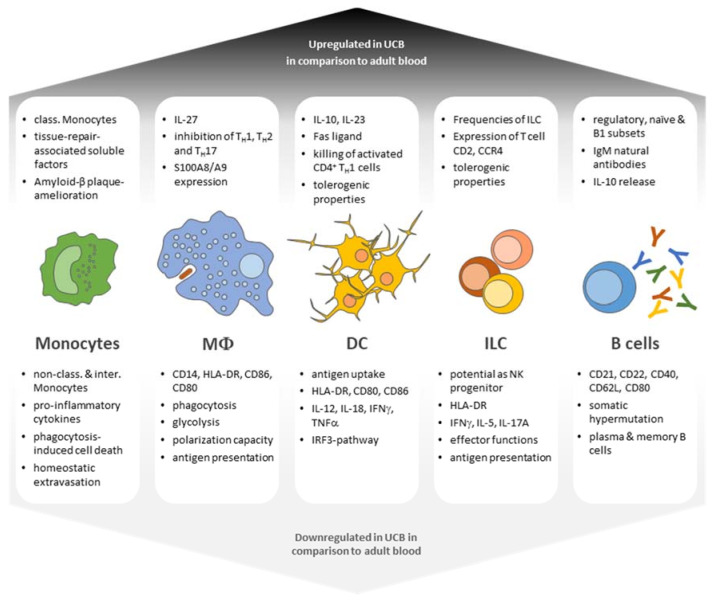
Functional characteristics of UCB leukocyte populations in comparison with adult leukocyte subsets. UBC-derived leukocyte subsets display both enhanced and diminished capabilities in terms of antigen responses as well as differentiation potential and generally present a phenotype of immaturity and tolerance (class. = classical; non-class. = non-classical; interm. = intermediate).

**Table 1 ijms-22-05923-t001:** Frequencies and total amounts of APC subpopulations in neonatal and adult blood.

Cell Subset	Adult		Cord Blood		References
	%	×10^3^/mL	%	×10^3^/mL	
Monocytes	5.28	397.85	7.91	1338.42	[14]
Classical	74.70	302.98	85.55	1160.95	
Intermediate	3.41	14.16	3.22	45.4	
Non-classical	12.00	43.33	4.81	61.25	
Dendritic cells	0.49	37.12	0.35	56.54	[14,33,34]
mDC	0.14	10.24	0.06	9.49	
pDC	0.05	3.55	0.06	8.14	
B cells	2.41	173.99	3.10	518.94	[14]
Immature	2.10	2.58	9.24	34.15	
Memory	22.5	31.39	-	-	
Plasmablast	2.60	4.6	-	-	
B1 cells	3.51	6.11	1.72	9.11	
ILC					[35]
ILC1	29.00	0.468	[35]	[35]	
ILC2	31.00	0.585	[35]	[35]	
ILC3	32.00	0.513	[35]	[35]	

Reference values of adult and neonatal blood APC subsets. Frequencies of monocytes, B cells, and DCDC are presented as frequencies of total leukocytes (CD45^+^). ILC frequencies are presented as frequencies of Lin^−^ CD27^+^. Subpopulations of APC are displayed as frequencies of the respective APC subset. Frequencies are displayed as medians.

**Table 2 ijms-22-05923-t002:** Pre-clinical and clinical trials using UCB APC for regenerative therapy and immunomodulation in mice and humans.

Condition	Recipient	UCB-APC	Effect	References
Acute ischemic stroke	Human	Monocytes	Ongoing recruitment	NCT02433509
middle cerebral artery occlusion	Rat	CD14^+^ monocytes	- increased release of neuroprotective soluble factors - reduced glial activation following oxygen and glucose deprivation	[156]
Cerebral palsy	Mouse	CD14^+^ monocytes	- significant reduction in infarct size and motor deficits - reduced CD4^+^ T cell infiltration and cortical cell death	[160]
Alzheimer’s disease	Mouse	CD14^+^ monocytes	- recovery of cognitive impairments - reduced amyloid plaque pathology	[161,162]
Pediatric AML	In vitro	CD34^+^-derived DC electroporated with WT1 mRNA	- potent WT1 presentation - induction of primary pediatric AML cells	[163,164]
Adult AML	In vitro and Mouse	CD34^+^-derived DC	- effective generation of Langerhans cells - induction of T_H_1 responses	[165]
Central nervous system injuries	Mouse	DUOC-01 (CD14^+^ monocytes)	- Remyelination in the corpus callosum - Reduced glial cell infiltration	[166,167]
Other	Ex vivo	CD34^+^-derived DC	- significant increase in CIK cell expansion - higher secretion rates of IL-12, IFNγ, and TNFα - similar cytotoxicity to standard CIK	[168]
	In vitro	Monocyte-derived DC exosomes	- Induction of allogenic T cell proliferation - greater cytotoxic activity against tumor cell lines	[169]
	In vitro	MoDC	- effective induction of allogenic cytotoxic Vγ9γδ T cell proliferation	[72,170]
			- enhanced killing of A549 cancer cells	

WT1 = Wilms tumor antigen 1, AML = acute myeloid leukemia, CIK = cytokine-induced killer cells.

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
