# Peer review of "Cord-Blood-Derived Professional Antigen-Presenting Cells: Functions and Applications in Current and Prospective Cell Therapies"

_ijms, 2021, doi:10.3390/ijms22115923_

Round 1

Reviewer 1 Report

This comprehensive review is carefully written but, in some specific parts, tedious to read. The authors should consider narrowing down the topic to make the review more detailed and more attractive to the readers. For example, the section describing the characteristics of the APC and antigen-presenting populations is too long and repetitive. I would suggest making it easier to read and focus only on the main differences between adults and neonatal cells. The title of the manuscript mentions the applications of these types of cells. I would expand this section, which will be for sure more appealing. 

The conclusion section is relatively superficial. The authors should expand it, discussing, for example, the main pro and cons of the different strategies with each subpopulation of UBC.

English style is ok. Check typos and double spaces in several points of the manuscript.

Reviewer 2 Report

Cunningham and Hackstein offer an expert and detailed review on PBMC profile and activities, comparing adult onset with perinatal tissue (umbilical cord blood). The manuscript is extremely detailed and updated. Of particular interest and instrumental to support neophytes in cell-based therapies for hematological treatments.

Unfortunately, we found the elaborate sometimes confused and confusing. Several sentences and paragraphs may benefit from revision or rephrasing. Visal support is limited to one figure (diagram) only. Additional Tables and diagrams may largely benefit the high-quality review and eventually relieve the crowded text from basic notions

In the neonatal paragraphs, there are frequently additional information or comparative description including important info about adult cells. Maybe a large unique paragraph can be considered, subdivided into secretive and cell-to-cell interactive activities. Just a suggestion to increase fluidity in the manuscript

 Several major and minor details need to be corrected before publication:

(line 58) Define IL-12p70 and IL10bla

(line 67) classical monocytes generate DC post-exposure to IL-4 and GM-CSF, while classical and non-classical fail to do so. Please correct and/or elaborate

UCB higher frequency in 3 different classes of mono. Please mention relative amount and percentages

(line105) why TNFalpha is within parenthesis?

Please mention and clarify role of non-polymorphic HLA (i.e. HLA-G) in M2 subclass formation

(line 173) gamma is missing in IFN word.

Similarly in lines 300, 302, 312, 388, 393, 647 there is an extra space suggesting a missing symbol/letter. Please correct
